# Production of Hybrid Nanocomposites Based on Iron Waste Reinforced with Niobium Carbide/Granite Nanoparticles with Outstanding Strength and Wear Resistance for Use in Industrial Applications

**DOI:** 10.3390/nano13030537

**Published:** 2023-01-28

**Authors:** Shams A. M. Issa, Abeer M. Almutairi, Karma Albalawi, Ohoud K. Dakhilallah, Hesham M. H. Zakaly, Antoaneta Ene, Dalia E. Abulyazied, Sahar M. Ahmed, Rasha A. Youness, Mohammed A. Taha

**Affiliations:** 1Department of Physics, Faculty of Science, University of Tabuk, Tabuk 47512, Saudi Arabia; 2Faculty of Science, Al-Azhar University, Assiut Branch, Assiut 71524, Egypt; 3Department of Chemistry, Faculty of Science, University of Tabuk, Tabuk 47512, Saudi Arabia; 4Institute of Physics and Technology, Ural Federal University, 620002 Yekaterinburg, Russia; 5INPOLDE Research Center, Department of Chemistry, Physics and Environment, Faculty of Sciences and Environment, Dunarea de Jos University of Galati, 800008 Galati, Romania; 6Department of Petrochemical, Egyptian Petroleum Research Institute (EPRI), Cairo 8575, Egypt; 7Surfactant Lab, Petrochemical Department, Egyptian Petroleum Research Institute (EPRI), Cairo 8575, Egypt; 8Spectroscopy Department, National Research Centre, El Buhouth St., Dokki, Giza 12622, Egypt; 9Solid State Physics Department, National Research Centre, El Buhouth St., Dokki, Giza 12622, Egypt

**Keywords:** recycling waste materials, iron, strength, CTE, wear resistance, industrial applications

## Abstract

The main objective of this work is to recycle unwanted industrial waste in order to produce innovative nanocomposites with improved mechanical, tribological, and thermal properties for use in various industrial purposes. In this context, powder metallurgy (PM) technique was used to fabricate iron (Fe)/copper (Cu)/niobium carbide (NbC)/granite nanocomposites having outstanding mechanical, wear and thermal properties. Transmission electron microscopy (TEM) and X-ray diffraction (XRD) examinations were used to investigate the particle size, crystal size, and phase composition of the milled samples. Additionally, it was investigated how different volume percentages of the NbC and granite affected the sintered specimens in terms of density, microstructure, mechanical and wear properties, and coefficient of thermal expansion (CTE). According to the findings, the milled powders included particles that were around 55 nm in size and clearly contained agglomerates. The results showed that the addition of 4 vol.% NbC and 8 vol.% granite nanoparticles caused a reduction in the Fe–Cu alloy matrix particle sizes up to 47.8 nm and served as a barrier to the migration of dislocations. In addition, the successive increase in the hybrid concentrations led to a significant decrease in the crystal size of the samples prepared as follows: 29.73, 27.58, 22.69, 19.95 and 15.8 nm. Furthermore, compared with the base Fe–Cu alloy, the nanocomposite having 12 vol.% of hybrid reinforcement demonstrated a significant improvement in the microhardness, ultimate strength, Young’s modulus, longitudinal modulus, shear modulus, bulk modulus, CTE and wear rate by 94.3, 96.4, 61.1, 78.2, 57.1, 73.6, 25.6 and 61.9%, respectively. This indicates that both NbC and granite can actually act as excellent reinforcements in the Fe alloy.

## 1. Introduction

The development of substantial research in the automobile brake friction sector over the last few decades has been driven by the growing customer demand for low-cost, high-performing, and environmentally friendly materials with a favorable price/quality ratio. Unfortunately, this goal is sometimes not possible and is difficult to achieve. For example, the major factor in the shutdown of wind turbines, and the cause of 40% of all wind turbine failures, is gearbox failure [1]. Gear wear is the main factor in gearbox failure due to complex operating conditions and heavy loads that the gears have to bear. Based on this fact, it is necessary to improve gear hardness and wear resistance with the aim of increasing the lifespan of gearboxes and thus enhancing the overall power generation efficiency of wind turbines. In this regard, ferro-composites (composites having an iron (Fe)-based matrix) are among the most suitable materials candidates for use in such applications [2,3,4].

Fe could be alloyed with copper (Cu) [5], cobalt (Co) [6,7], tin (Sn) [8], nickel (Ni) [9], Cu–Co [10], Cu–Ni [11,12], etc., for improving resistance of the metallic matrix. It should be noted that Cu and its alloys are primarily preferred because they increase thermal conductivity in the frictional contact and maintain the level of coefficient of friction at high temperatures by forming Cu oxides there [1,10]. Moreover, adding Cu to it has an economical dimension, as it reduces the temperature required for sintering during preparation. This can be attributed to that the melting point of Cu (~1080 °C) is lower than that of Fe (~1538 °C) [1,13,14]. It is worth noting that there are many ceramics that can be used as a reinforcer for metal matrices, including titanium (Ti), titanium carbide (TiC), carbon nanotube (CNT), titanium diborate (TiB_2_), boron carbide (B_4_C), alumina (Al_2_O_3_), tungsten carbide (WC), molybdenum disulfide (MoS_2_) and yttrium oxide (Y_2_O_3_) are used as reinforcement materials [15].

Niobium carbide (NbC) has drawn a lot of study interest due to its exceptional combination of great chemical stability, low electrical resistance, high melting point (nearly 3610 °C), excellent thermal stability, and enormous hardness. Its usage as a reinforcing component in the matrix of Fe and its alloys to improve their mechanical characteristics and wear resistance is the most significant of its numerous technical uses [16,17,18]. Additionally, granite dust, a ceramic waste made from granite rocks, is a promising material for use as a reinforcement to increase the above properties of Fe and its alloys. It was chosen to perform this task due to its high level of hardness, strength, and modulus, as well as its low cost and low level of weight. Notably, granite dust has a composition that is primarily made up of SiO_2_ and Al_2_O_3_ [19,20].

Powder metallurgy (PM) is a cost-effective method for the mass production of metal and its alloy matrix composites. The reason for this is their ability to combine the intense refinement of beginning powders with the homogeneous dispersion of the reinforcement [21,22,23]. The objective is to blend the various characteristics of the two or more materials to create a brand-new material with superior attributes. However, one of the challenging tasks for the researchers using this method is to improve the interphase’s bonding strength by managing the manufacturing process and a few particle characteristics, such as size, shape, and volume percentage. PM provides a number of important benefits, such as improved control over reinforcement distribution and creation of a more uniform matrix microstructure, which decreases segregations [24]. A stronger mechanical bonding between the components is also guaranteed by PM because of the high energy involved in impact events between the milling medium and the mixture powder [25,26].

Although Fe is important in the manufacture of gears, brakes, etc., its poor mechanical properties and wear resistance limit its uses in these applications, or at least reduces its life span. Therefore, the major goal of this work is to manufacture inexpensive Fe–Cu alloy matrix nanocomposites complemented by hybrid NbC and granite wastes using Fe waste by a PM approach. This objective was achieved by collecting leftover supplies of Fe from previous industrial processes and then grinding it into powders with the help of a high-energy ball mill; then, preparing Fe-10 vol.% Cu matrix nanocomposites reinforced by hybrid NbC and granite nanoparticles with varying volume percentages. The prepared composites were subjected to a sintering process at 1150 °C for 1 h in argon atmosphere. Finally, the physical and tribo-mechanical properties of the sintered nanocomposites were measured. Since these nanocomposites experience high temperatures during fraction, we also seek to reduce the CTE so that they can be widely used in previous industrial applications. 

## 2. Experimental Set Up

### 2.1. Samples Preparation

Fe waste resulting from the remnants of the industries of some parts of spare parts and other Fe supplies in the lathe workshops or Fe factories are reused again. Moreover, NbC and granite with particle sizes of about 45 and 47 nm, respectively, were used as reinforcements. Fe waste was first broken up into little pieces that were a few millimeters wide, and then it was ground into waste-powder for three hours in a high-energy ball mill (HEBM). Subsequently, varied volume percentages of the hybrid NbC phase and granite waste were introduced to the Fe-10 Cu alloy, which was selected to function as the matrix. Table 1 and Table 2 provide the chemical compositions of Fe and granite wastes, respectively, while Table 3 lists the compositions of the batches intended for nanocomposites samples along with their acronyms. Each specimen was ground for 10 h at 550 rpm with a ratio of balls to powder = 20:1. The nanocomposite powder was then sintered in argon gas for 1 h at 1150 °C with a heating rate of 5 °C/min; after that, pressing with a hydraulic machine with pressure of 40 × 10^6^ Pa.

### 2.2. Description of Starting Materials and Milled Nanocomposites Powders 

X-ray diffraction (XRD; Philips PW 1373, Philips, Amsterdam, Netherlands) technique was used to analyze the phase composition of the stating materials and the prepared samples. Furthermore, with the aid of our most recent papers [27], the size of crystal, strain of lattice, and density of dislocation were determined from the XRD pattern. The average particle size of the raw materials and milled powders has been measured using a transmission electron microscopy (TEM, Oxford, Abingdon, UK). Field emission scanning electron microscopy (FESEM, PhotoMetrics, Huntington Beach, CA, USA) was utilized to study the morphology of the sintered samples.

### 2.3. Properties of the Sintered Nanocomposites

#### 2.3.1. Physical Properties

Each sintered samples’ density (*ρ*) and apparent porosity (A.P.) were calculated using the formulas after being measured using the Archimedes technique [28]:(1)$$\rho =\left(w_{s}-w_{d}\right)/\left(w_{s}-w_{i}\right)$$
(2)$$A.P.=\frac{W_{d}}{W_{s}-W_{i}}\times \rho_{l}$$

The theoretical density (Th.D.) of the samples was calculated using a mixture rule as per the formulae below:(3)$$\mathrm{Th}.D.=\rho_{m}V_{m}+\rho_{N}V_{N}+\rho_{G}V_{m}$$
where *ρ_m_*, *ρ_N_* and *ρ_G_* are the density value of Fe alloy, NbC and granite, respectively, while V_m_, V_N_ and V_G_ are the volume percent of Fe alloy, NbC and granite, respectively.

The relative density (R.D.) was calculated using the values of bulk and theoretical densities, in the formulae below:(4)$$\mathrm{RD}=\frac{B.D}{\mathrm{Th}.D}\times 100$$

#### 2.3.2. Thermal Analysis 

Thermal expansion measurements of the sintered Fe alloy-based hybrid nanocomposites samples were studied from 25 to 1000 °C in air and CTE values calculated were investigated.

#### 2.3.3. Mechanical Properties

A Vickers tester was used to quantify the microhardness of the sintered nanocomposites with 1.9 stress for 20 s, as was indicated in our most recent study [29]. Moreover, ASTM E9–19 was used to evaluate the ultimate strength of the sintered samples.

The values of the elastic moduli: longitudinal modulus (*L*), shear modulus (*G*), Young’s modulus (*E*), bulk modulus (*B*), and Poisson’s ratio (*ν*) were determined using the pulse-echo method, the longitudinal (V_L_) and shear (V_S_) ultrasonic velocities of the nanocomposites [30,31]. 

#### 2.3.4. Tribology Test

Through the utilization of a pin-on-disk wear equipment and samples that wear 1.5 and 0.6 cm in diameter and thickness, respectively, on a pin-on-disk wear tester, dry sliding wear tests were performed in the air simulating sliding conditions. Two applied loads (10 and 30 N), various sliding distances (300, 700, 1100 and 1500 m), and a 0.8 m/s speed made up the process conditions for the wear test [26].

## 3. Results and Discussions

### 3.1. XRD Analyses

Photographs of Fe waste taken before and after 3 h of HEBM milling are depicted in Figure 1a,b. It should be obvious that the Fe waste material is gray in color and appears as little crusts before grinding (Figure 1a). Fe is ground until it is very fine and turns into a dark grey powder (Figure 1b).

The XRD patterns of the raw materials, i.e., Fe, Cu, NbC and granite powders, are shown in Figure 2a,b. According to (ICCD file cards: 89-4185, 85-1326 and 89-3830), respectively, the patterns show the existence of peaks corresponding to Fe, Cu, NbC, and granite. Since granite consists of three phases; namely quartz (SiO_2_), albite (Na(AlSi_3_O_8_)) and annite (KFe_3_^2+^AlSi_3_O_10_(OH))_2_, it should be identified according to the presence of the characteristic XRD peaks of these components (ICCD file cards 88-2487, 89-6430 and 42-1413, respectively).

The series of XRD spectra of the Fe–Cu alloy-based nanocomposites having different volume percentages of hybrid reinforcements (i.e., NbC and granite) after milling for 10 h are shown in Figure 3. According to the existence of their distinctive XRD peaks, it appears that only Fe and Cu phases are seen in the FNG0, FNG1 and FNG2 samples. For the FNG4 sample, the characteristic XRD peaks for granite are clearly visible next to those represented by Fe and Cu. However, the characteristic XRD peaks of NbC are seen only in the last sample, i.e., FNG8. The main reason behind the absence of NbC and granite peaks in FNG1 and FNG2 samples and the absence of XRD peaks of NbC in the FNG4 sample is that the levels of these reinforcements are minimal and regarded as being below the XRD device’s detection limit [32]. It is also observed with the increase of the contents of the reinforcements in the nanocomposite samples, the width of the Fe and Cu peaks increases and the intensity of the same peaks also decreases. The size of the crystal, density of the dislocation, and lattice strain of the strong peaks for all samples were estimated and summarized in Table 4 in accordance with these observed changes. This table shows how the concentration of hybrid ceramics causes a decrease in crystal size and an increase in lattice strain and dislocation density. These outcomes can be attributed to the significant lattice deformation and grain size refinement that take place during the milling process [33,34]. XRD charts specify crystal sizes of 29.73, 27.58, 22.69, 19.95, and 15.80 nm for the ground powders of specimens labeled FNG0, FNG1, FNG2, FNG4, and FNG8, respectively.

### 3.2. TEM Observations

The TEM images of the hybrid reinforcements (i.e., NbC and granite) used and the powders of nanocomposites; namely FNG0, FNG4 and FNG8 after 10 h of milling, are represented, respectively, in Figure 4a,b and Figure 5a–c, respectively. It is obvious from Figure 4 that the NbC and granite particles are spherical in shape and have sizes of about 45 and 47 nm, respectively. In addition, the average particle size of the Fe–Cu alloy matrix (FNG0 sample) powder was 87.48 nm with a noticeable agglomeration. However, after adding 6 and 12 volume percentages of the hybrid reinforcement, the particle size decreased significantly, i.e., 61.30 nm (FNG4) and 32.46 nm (FNG8). Another important observation in these images is that this successive addition of the reinforcement results in less agglomeration of the particles. Generally, during mechanical alloying, the powder particles are subject to repeated plastic deformation, cold-welding and fracture. On the other side, the cold-welding causes increase in the agglomerate particle, while the fracture is leading to the decrease in particle sizes. Consequently, the Fe–Cu alloy matrix (ductile) particles undergo deformation while ceramics reinforcements such as NbC and granite (brittle) particles undergo fragmentation. As a result, when balls collide, the Fe–Cu alloy particles begin to weld and the ceramics particles come between two or more matrix particles. Thus, the ceramics particles remain at the boundaries of the welded Fe–Cu alloy particles and, consequently, the nanocomposite powders are prepared with a decrease in the size of particles. It is possible to explain this decrease in the particle size and agglomeration of the Fe–Cu alloy matrix with the increase in the content of the hybrid ceramics with the knowledge that the increase in the percentage of ceramic leads to an increase in the local deformation of the matrix, which accelerates the rate of work hardening of the Fe–Cu matrix [35]. These results are in good agreement with the works [22,36,37]. Figure 6 illustrates the impact of the hybrid reinforcement on the Fe alloy matrix’s particle size.

### 3.3. Characterization of the Sintered Samples by SEM 

SEM images of the FNG0, FNG1 FNG2, FNG4 and FNG8 specimens sintered at 1150 °C for one hour in argon atmosphere are displayed in Figure 7a–e. The unreinforced Fe alloy (FNG0) appeared to have acceptable densification, as shown by significant grain growth consistent with the existence of the few pores. This observation proves that the selected temperature was suitable for sintering Fe–Cu alloy. The contact between the matrix grains is noticeably weaker as a result of the presence of ceramic reinforcement particles, which are known to have been discovered near the border of the Fe alloy matrix grains. In addition to this, it was found that the amount of hybrid reinforcement particles in the investigated nanocomposite samples had an effect on the porosity of the samples. This is consistent with results for porosity which will be discussed in the following section. 

### 3.4. Physical Parameters

Density is known to be a very important property to be measured for composites prepared by PM. It is common knowledge that the mechanical characteristics of nanocomposites are changed during the sintering process by the pores that may form as a result of the necking of the powder particles. The bulk density, relative density, and apparent porosity of the Fe alloy and its nanocomposites, sintered at 1150 °C for 1 h in an environment of argon, are shown in Figure 8a–c. The theoretical densities of the FNG0, FNG1 FNG2, FNG4, and FNG8 samples, which are 7.791, 7.971, 7.863, 7.755 and 7.539 g/cm^3^, respectively, are taken into consideration. It can be seen from this graph that the increasing volume percentage of the hybrid reinforcement is what causes the samples’ notable decrease in bulk and relative densities and rise in porosity values. This decrease in both ρ and RD with an increase in apparent porosity is due to several factors. First of all, the density of the granite reinforcement, 2.65 g/cm^3^, is significantly lower than that of the Fe–Cu matrix, 7.97 g/cm^3^. Second, less particle rearrangement occurs during sintering because the melting point of NbC (~3490 °C) is substantially higher than that of Fe (~1.538 °C) and Cu (~1083 °C). Finally, the volume % of additive hybrid ceramics has a direct impact on the growth of the contact surfaces, the production of closed pores, and the growth of the grain.

### 3.5. Thermal Expansion

The CTE values and relative thermal expansion (dL/L) of the FNG0, FNG1 FNG2, FNG4, and FNG8 samples sintered for 1 h at 1150 °C measured at temperatures between 25 and 1000 °C as represented in Figure 9a,b. The dL/L of the sintered samples rose with rising temperature as was predicted. The results show that adding various hybrid reinforcement volume ratios causes the dL/L of the Fe alloy to steadily drop. For example, the dL/L value for FNG0, FNG1 FNG2, FNG4 and FNG8 samples measured at 100 °C is 2.001 × 10^−3^, 1.965 × 10^−3^, 1.689 × 10^−3^, 1.425 × 10^−3^ and 1.032 × 10^−3^, respectively. On the other hand, the values recorded for the same samples measured at 1000 °C are 13.242 × 10^−3^, 12.911 × 10^−3^, 12.216 × 10^−3^, 11.155 × 10^−3^, 9.331 × 10^−3^, respectively. As demonstrated in Figure 9b, the CTE values drop as the volume percentages of the hybrid reinforcements−NbC and granite increase. The CTE of a Fe alloy is 12.49 × 10^−6^/°C, but it drops to 12.12 × 10^−6^, 11.70 × 10^−6^, 10.85 × 10^−6^ and 9.25 × 10^−6^/°C after adding 1.5, 3, 6 and 12 vol.% of hybrid reinforcements, respectively. The thermal mismatch between the Fe alloy matrix and the reinforcements (NbC and granite) in the nanocomposite samples, which actively contribute to the knowledge of the dL/L behavior, may be the source of the observed reduction in the dL/L and CTE values. Additionally, the lower CTE values of NbC and granite reinforcement (≈6.7 × 10^−6^ and 7.8 × 10^-6^/°C) as compared to Fe–Cu alloy matrix (11.9 × 10^−6^ and 16.5 × 10^−6^/°C, respectively) [38]. Therefore, adding ceramics to the Fe–Cu alloy leads to a decrease in the CTE value. The linear reduction in CTE values caused by the addition of hybrid ceramic particles produces thermally stable nanocomposites and emphasizes the significant contribution of NbC and granite particles to the enhancement of Fe alloy dimensional stability [39].

Since there are no previous studies on the CTE of Fe and its alloys, but there are studies on the improvement of CTE of metal alloys. For example, Reddy et al. [39], Moustafa et al. [40], and Moustafa et al. [38] emphasized that reinforcing particles such as SiC, yttrium–silica fume, TiB_2_ and tantalum carbide (TaC)–NbC have a significant effect on remarkable improving CTE of Al, Mg10Li5Al Alloy, Al–Mg alloy, Al–Cu alloy, respectively.

### 3.6. Mechanical Properties

Several factors, including particle shape, size, amount, distribution, density of the reinforcement, and preparation method, might affect the values of mechanical properties. It is crucial to keep in mind that microhardness is a highly beneficial characteristic since it illuminates the overall mechanical behavior of composites [41]. Due to this importance, the fluctuation in the microhardness values of the Fe–Cu alloy and its nanocomposites were recorded along with those of ultimate strength and are represented in Figure 10a,b. It has been noted that the nanocomposite materials microhardness ratings are greater than those of Fe alloy. It has been observed that the microhardness value of Fe alloy and nanocomposite with 1.5, 3, 6, and 12 vol.% of hybrid reinforcements are 138.88, 148.94, 166.34, 204.97 and 269.87 Hv, respectively. As indicated in Figure 11 and Figure 12, other mechanical properties of the samples at the same sintering temperatures in terms of ultrasonic velocities and group of elastic moduli were also examined using an ultrasonic non-destructive approach. The findings demonstrate that as hybrid reinforcement is increased, the elastic group moduli and ultrasonic (i.e., longitudinal and shear) velocities rise. As the proportion of hybrid reinforcement gradually increases from 0 to 12 vol.%, the longitudinal velocity value ranges from 5457.21 to 7298.81 m/s and the shear velocity value ranges from 2962.25 to 3841.08 m/s. Moreover, Young’s modulus value for FNG0 sample (zero reinforcement) is 180.61 Gpa, while for samples FNG1, FNG2, FNG4 and FNG8, it increased to 188.44, 207.73, 237.04 and 291.08 GPa, respectively. 

Numerous elements, which may be described as follows, contributed to the extraordinary improvement in mechanical characteristics, such as microhardness, strength, and elastic moduli.
According to Equation (5), the inclusion of uniformly distributed NbC and granite reinforcements can increase the microhardness of nanocomposites [42].
(5)$$H_{\mathrm{Nanocomposite}}=H_{Fe\;alloy}V_{Fe\;alloy}=H_{NbC}V_{NbC}+H_{granite}V_{granite}$$
It is known that the microhardness (H) and the rest of the mechanical properties have a strong relationship with the grain size (d) based on the Hall–Petch effect, as shown in Equation (15) [43]. The addition of ceramics increases the dislocation density as well as grain refinement, which leads to an increase in the grain boundaries. Thus, the dislocation movement is hindered, which leads to an improvement in strength and other mechanical properties [44].
(6)H=H0Kd−12
where H_0_ and K are the constant.
The Orowan mechanism states that the strength of composite materials results from the interaction between reinforcing particles and dislocations. After dislocations travel through them, the remaining dislocation loops are positioned around each particle. In fact, these particles increase the material’s strength by preventing the movement of dislocations [45].The interfacial contact between the Fe alloy and the ceramics particles is sufficient, as evidenced by the effective load transfer (σ_efficient_) from the Fe alloy matrix to the NbC and granite reinforcement during compressive testing, which helps to strengthen the nanocomposites.
(7)$$\sigma_{efficient\;}=0.5\;V\sigma_{F}$$
where *V* represents the amount of hybrid reinforcements as a percentage of volume and *σ_F_* is the yield strength of Fe alloy. The effect of reinforcement contents on the mechanical properties of Fe and its alloys matrix composites has been studied previously. Among them, Khazaal et al. [46] prepared Fe matrix hybrid composites reinforced with 0, 1, 2, 3 and 4 vol.% mixture (80% Al_2_O_3_ and 20% ZrO_2_) using PM. As indicated by the results obtained, the mechanical properties including microhardness and compressive strength were significantly improved by 37 and 71%, respectively, after adding 3 vol.% of the selected reinforcement. Raghav et al. [6] investigated the influence of adding chromium (Cr) particles on the microhardness and compressive strength behavior of a Fe-cobalt (Co) matrix. The results showed that the microhardness and compressive strength improved by about 17 and 20% after adding 10 wt.% of Cr particles. Kheradmand et al. [47] studied the effect of different SiC volume ratios up to 10 vol.% on the mechanical properties of Fe–Cu/graphite–SiC-BaSO_4_ hybrid composites. The results showed a remarkable improvement in the mechanical properties by further adding SiC particles. 

### 3.7. Wear Analysis 

With a variety of weights being applied; namely 10 and 30 N, the weight loss and wear rate variation of the Fe alloy and its composites are illustrated in Figure 13 a,b and Figure 14 a,b. It is evident that, for a given applied force, weight loss and wear rate increased almost linearly with an increase in the distance traveled by the sliding object. However, as the volume fraction of the hybrid reinforcements increased, the resistance to abrasive wear decreased significantly. Additionally, it was found that when the applied load rose to its maximum value, the effect of the sliding distance intensified.

For the sample FNG0, the weight loss is 8.34, 6.48, 3.87 and 2.75 mg for the sliding distance of 300, 700, 1100 and 1500 m, respectively, at an applied load of 10 N, while for the sample FNG8, its value is 3.47, 2.42, 1.51 and 1.05 mg, respectively, at the same applied load and distance. With the applied load increased to 30 N, the weight loss for the FNG0 sample at the same distance is 10.75, 7.96, 5.29 and 3.55 mg, respectively, and for the FNG8 sample is 4.76, 3.12, 2.47 and 1.32 mg, respectively. This could be related to the heat produced when the two abrasive mating surfaces were involved in the abrasive wear. The softening of the samples as a result of the resulting heat increases with the increase in the sliding distance. Because of this diminished bonding between the ceramics reinforced particles (NbC and granite) and the Fe alloy matrix as a result of microthermal softening of the Fe alloy matrix, the reinforcing particles were easily displaced during abrasion. In order to transfer stresses from the Fe alloy matrix to the ceramics particles and reduce sample wear, strong interfacial bonding is essential. As the applied load rises, it has been discovered that composite materials wear more quickly [48,49]. Conversely, weight loss, wear rate, and the impact of the applied load diminish as the volume% of hybrid reinforcement particles increases. For instance, at a sliding distance of 1500 m, the wear rate of the Fe alloy sample (FNG0) is 0.00459 and 0.0059 mg/s, respectively, for applied loads of 10 and 30 N. However, after adding 16 vol.% of hybrid reinforcement, the wear rate drops to 0.00175 and 0.00220 mg/s, respectively. This may be explained by the addition of NbC and granite particles to the Fe alloy, which increased microhardness and decreased actual area of contact while also increasing wear resistance. It is clear that reinforcement particles can serve as load-bearing components in nanocomposites, raising such nanocomposites wear resistance. Additionally, when the applied load increases, the pressure between the mating surfaces increases as well, causing a higher loss of weight and a faster rate of wear [49,50]. The highest heat generation in the two mating components and weakening of the interfacial connection between the matrix and reinforcement cause the biggest wear rate to occur at higher applied loads and maximum sliding distances. However, the influence of the applied load became more essential as the sliding distance grew. Açıkgöz et al. [51] investigated the wear behavior of the Fe-matrix composites reinforced by boron carbide (B_4_C) prepared by PM technique and they found that the wear resistance of the Fe-based composites was improved by 50% due to the addition of 5 wt.% B_4_C compared to the Fe matrix. Zhang et al. [52] improved the corrosion resistance of the Fe-Al-Cr alloy by 20% by adding 15 wt.% titanium carbide (TiC) obtained by hot-pressing sintering compared to the non-reinforced Fe alloy. 

Using FESEM measurements at various magnification powers, the worn surfaces of the FNG0 and FNG8 samples were examined at a sliding distance of 1500 m and under an applied stress of 30 N to evaluate the wear processes of the Fe–Cu matrix and its nanocomposites, as shown in Figure 15a–d. For the Fe–Cu matrix, only loose layers and grooves appear on the wear track as shown in Figure 15a,b. Surface delamination reveals adhesive wear, which includes crack initiation and propagation as well as final fracture of the material in the vicinity of the surface [53,54]. Figure 15c,d shows that the FNG8 sample has a smoother surface than the FNG0 sample, and there is only sporadic debris and slight grooves on the worn surface. Some Fe–Cu alloy debris has flattened in the wear process because of its low microhardness. Very low cracks appear in this wear track, thus, the dominant wear mechanism is abrasive wear [55,56].

## 4. Conclusions

The major goals of this research were to develop hybrid Fe-10 vol.% copper (Cu)-based nanocomposites reinforced with granite waste and niobium carbide (NbC) using powder metallurgy (PM) approach. The research results were as follows:It was discovered that the PM approach, in addition to having a high capacity to evenly distribute the hybrid reinforcement in the Fe alloy matrix and minimize the particle size of the milled powders, is ideal for obtaining Fe waste in powder form.The particle and crystallite sizes decreased with the increase of the ceramics content, reaching 38.25 and 17.82 nm for the sample that contained 4 vol.% NbC and 8 vol.% granite fume (FBG8).The bulk density and relative density of the Fe–Cu alloy matrix decreased gradually with the increase of the contents of hybrid reinforcements, while the apparent porosity increased.The coefficient of thermal expansion (CTE) of the Fe alloy was reduced with an increase in the percentage of the reinforcement particle size up to 9.25 × 10^−6^/°C for sample FNG8. In other words, the CTE of this sample decreased by about 26% compared to the non-hardened alloy (FNG0), which indicated high dimensional stability.The remarkable improvement in microhardness, ultimate strength and longitudinal modulus reached 269.9, 383.1 and 401.62 GPa for the sample, which improved about 93.7, 74.8 and 68.8% compared to the FNG0 sample.The Fe alloy’s wear resistance was considerably enhanced by introducing hybrid reinforcement and increased sliding slip, which decreased with increasing applied stresses. The wear rates for the FNG0 and FNG8 samples were 0.0222 and 0.0092 mg/s, respectively, after a sliding distance of 300 m and an applied force of 10 N, while the wear rates were 0.00138 and 0.00052 mg/s, respectively, after a sliding distance of 1500 m at the same applied load.Based on the promising results obtained, it can be concluded that the prepared nanocomposites can be useful for application in automobile brakes, gear boxes, and wind turbines.

## Figures and Tables

**Figure 1 nanomaterials-13-00537-f001:**
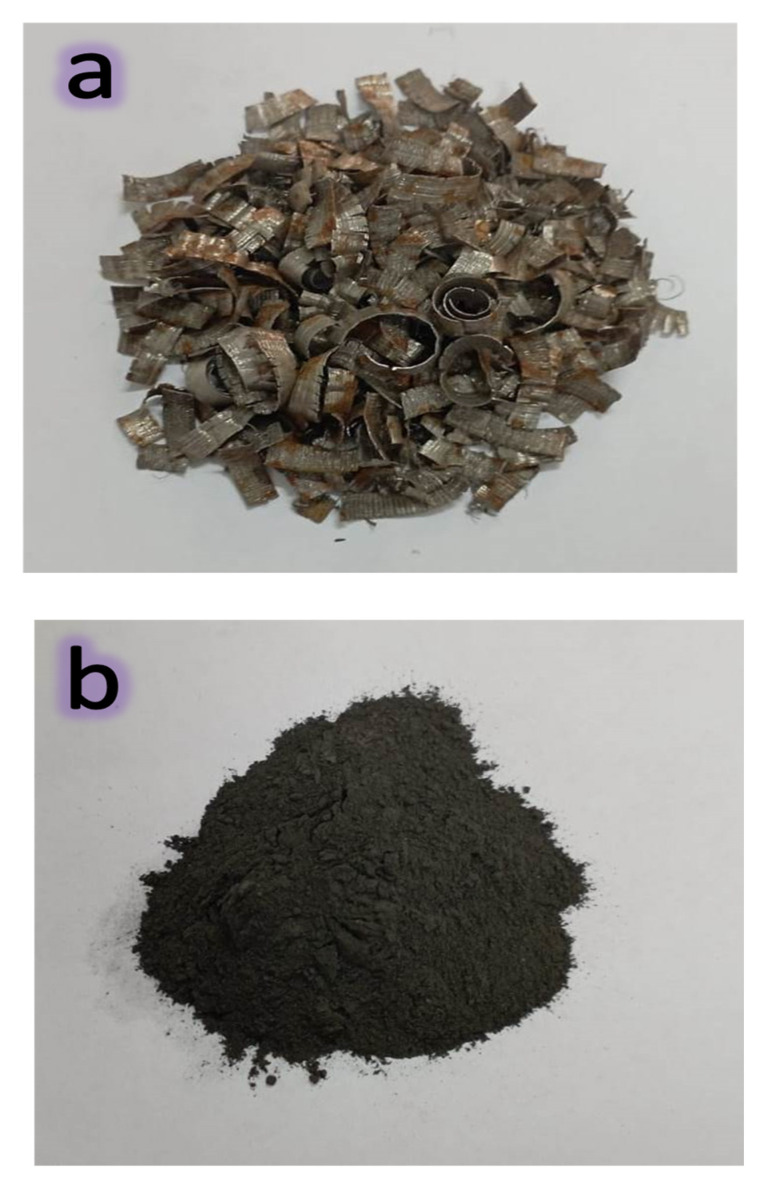
Photographs of Fe waste (**a**) before and (**b**) after milling for 3 h.

**Figure 2 nanomaterials-13-00537-f002:**
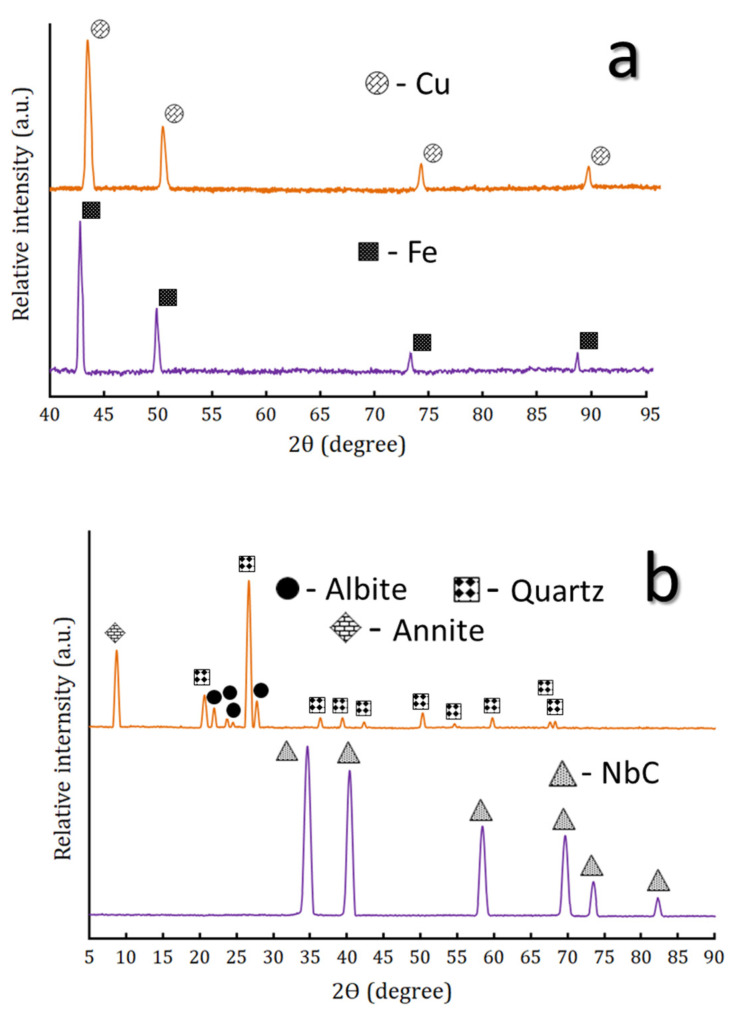
The XRD patterns of (**a**) as-received Fe waste after milling and Cu powders, and (**b**) as-received NbC and granite powders.

**Figure 3 nanomaterials-13-00537-f003:**
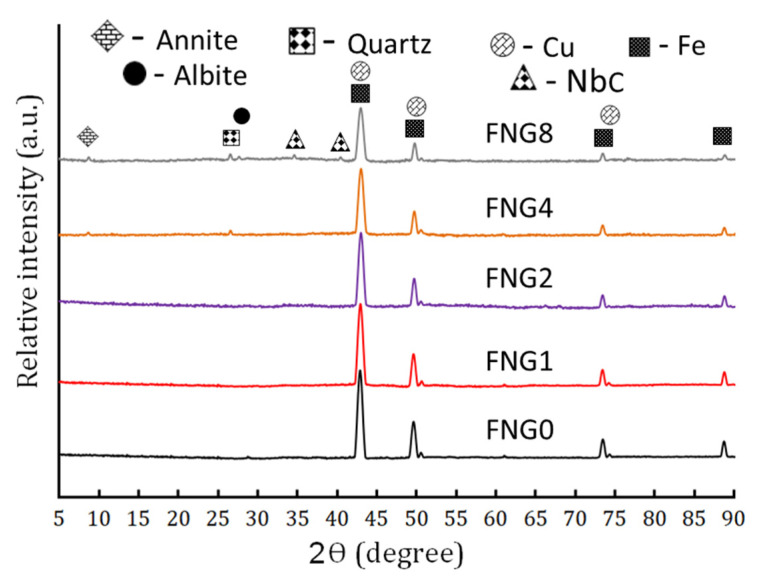
The XRD patterns of FNG0, FNG1, FNG2, FNG4 and FNG8 after 10 h of milling.

**Figure 4 nanomaterials-13-00537-f004:**
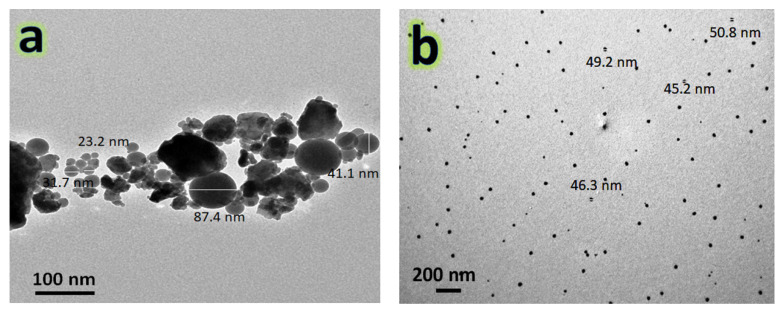
TEM micrographs of (**a**) NbC and (**b**) granite reinforcements.

**Figure 5 nanomaterials-13-00537-f005:**
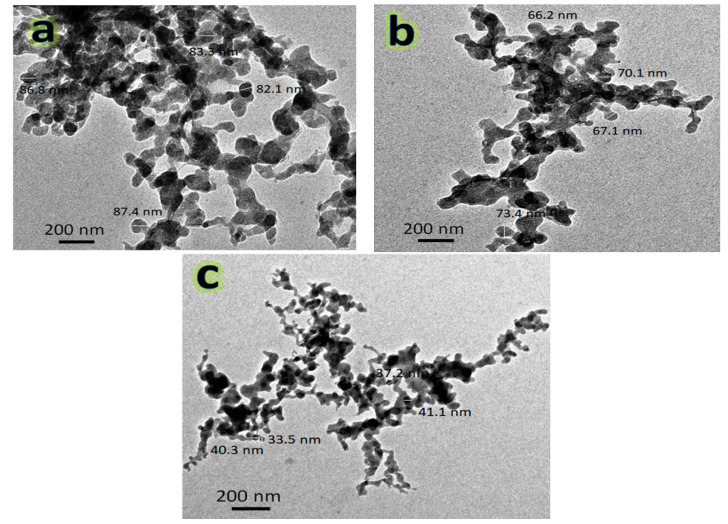
TEM micrographs of (**a**) FNG0, (**b**) FNG4, and (**c**) FNG8 nanocomposite powders after 10 h of milling.

**Figure 6 nanomaterials-13-00537-f006:**
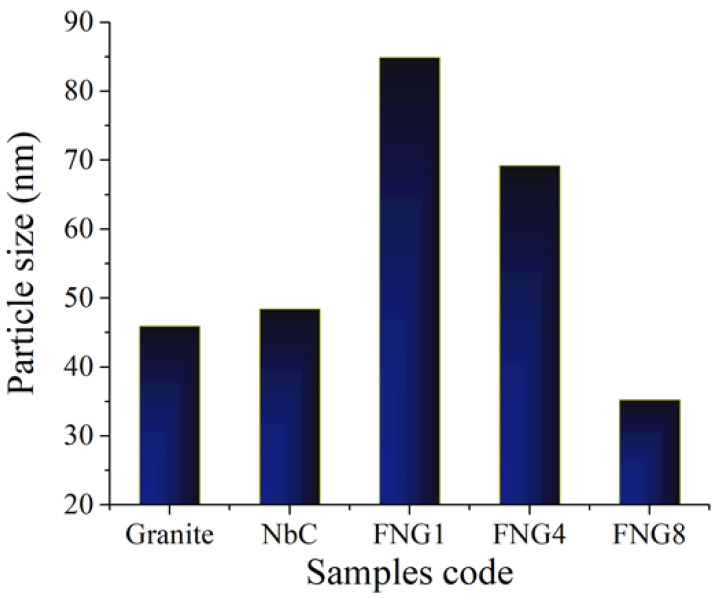
The effect of added hybrid reinforcements on the particle size of the Fe alloy matrix.

**Figure 7 nanomaterials-13-00537-f007:**
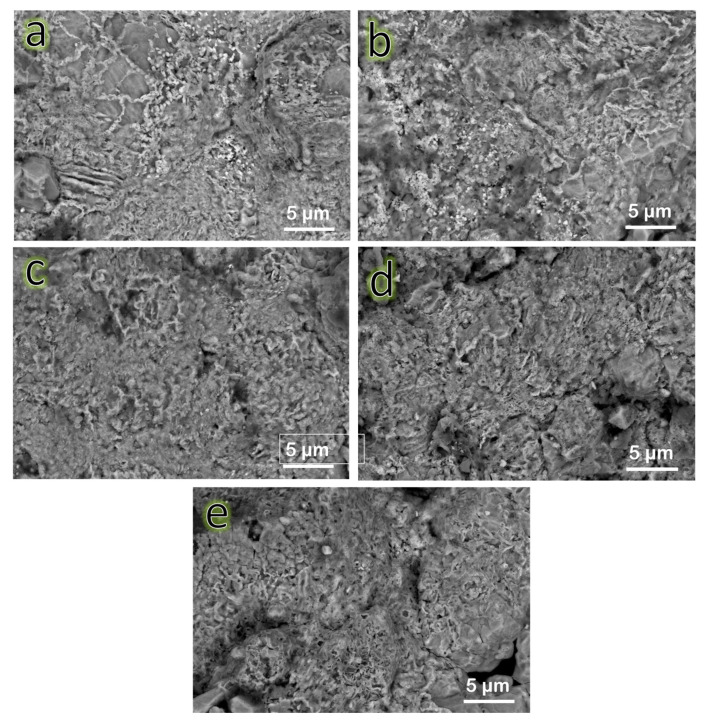
SEM images of (**a**) FNG0, (**b**) FNG1, (**c**) FNG2, (**d**) FNG4, and (**e**) FNG8 samples sintered at 1150 °C.

**Figure 8 nanomaterials-13-00537-f008:**
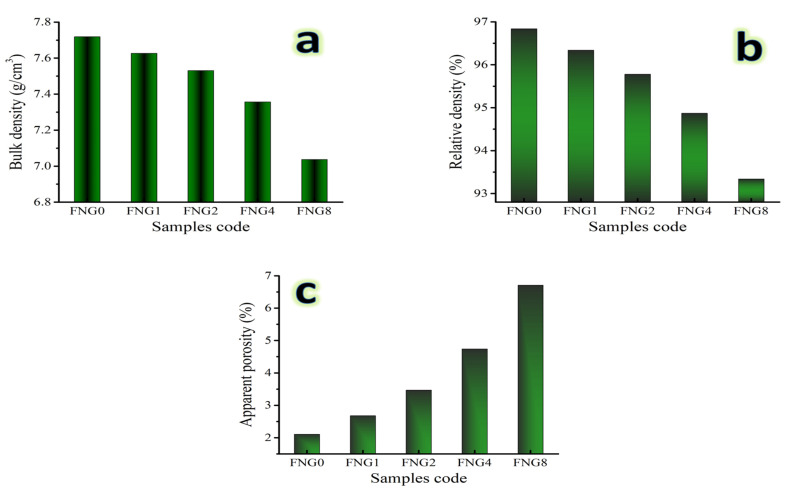
(**a**) Bulk density, (**b**) relative density, and (**c**) apparent porosity of the samples sintered for 1 h at 1150 °C.

**Figure 9 nanomaterials-13-00537-f009:**
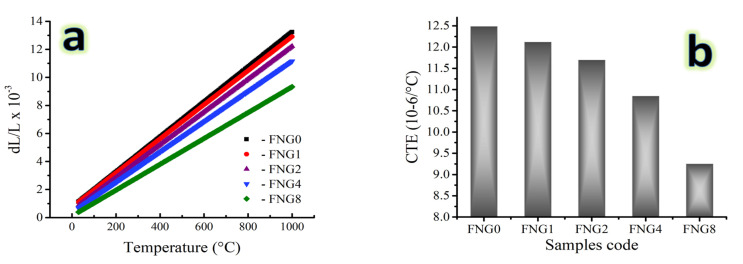
(**a**) Thermal expansion behavior and (**b**) CTE value of the sintered nanocomposites.

**Figure 10 nanomaterials-13-00537-f010:**
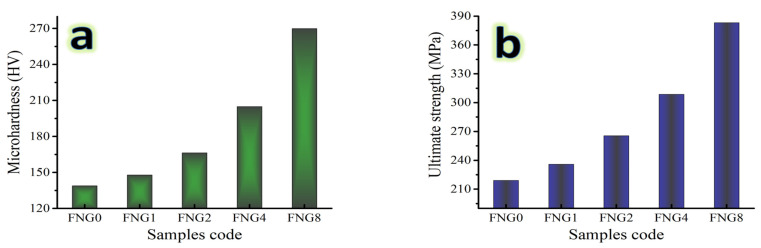
(**a**) Microhardness and (**b**) ultimate strength of sintered Fe–Cu alloy and its nanocomposites.

**Figure 11 nanomaterials-13-00537-f011:**
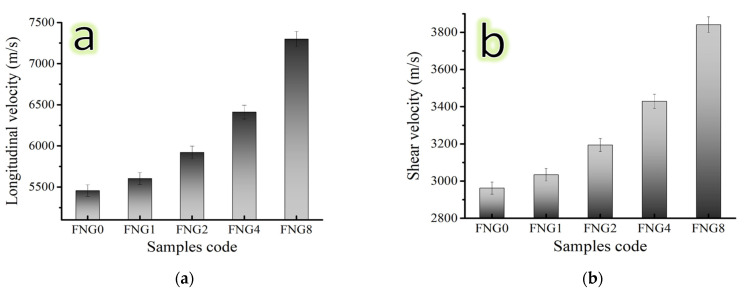
(**a**) Longitudinal and (**b**) shear velocity of sintered Fe–Cu alloy and its nanocomposites.

**Figure 12 nanomaterials-13-00537-f012:**
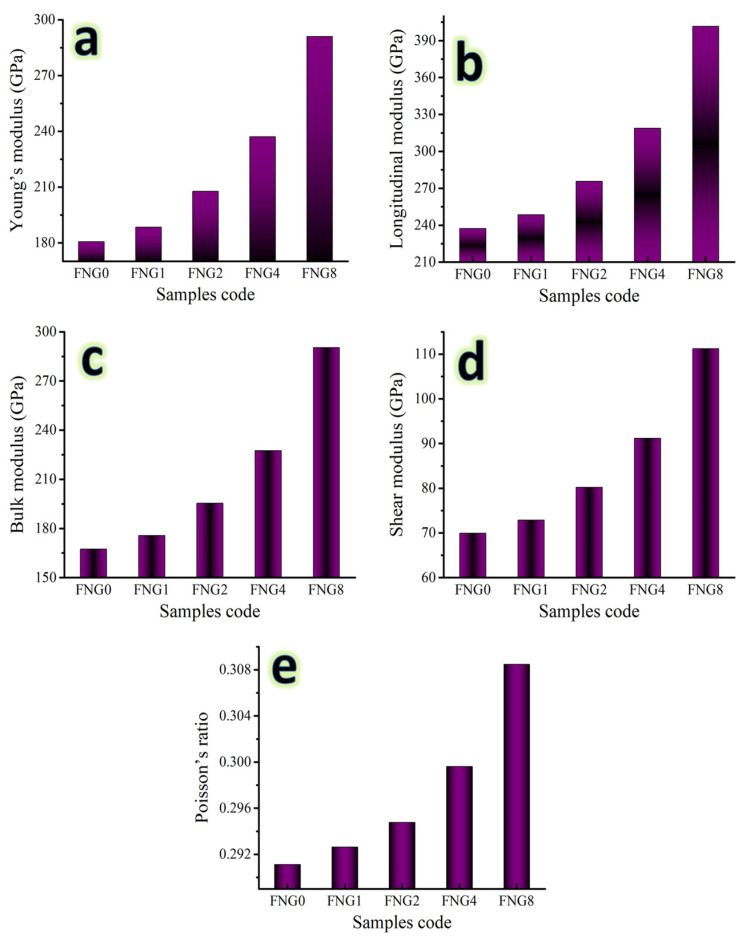
The group of elastic moduli of various sintered samples in terms of (**a**) Young’s modulus, (**b**) longitudinal modulus, (**c**) bulk modulus, (**d**) shear modulus and (**e**) Poisson’s ratio.

**Figure 13 nanomaterials-13-00537-f013:**
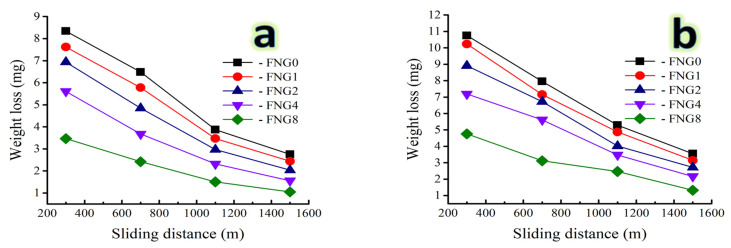
Weight loss of the Fe alloy and its nanocomposites as a function of the sliding distance at a load of (**a**) 10 N and (**b**) 30 N.

**Figure 14 nanomaterials-13-00537-f014:**
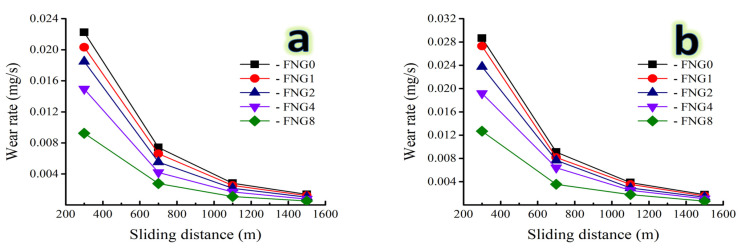
Wear rate of the Fe alloy and its nanocomposites as a function of the sliding distance at a load of (**a**) 10 N and (**b**) 30 N.

**Figure 15 nanomaterials-13-00537-f015:**
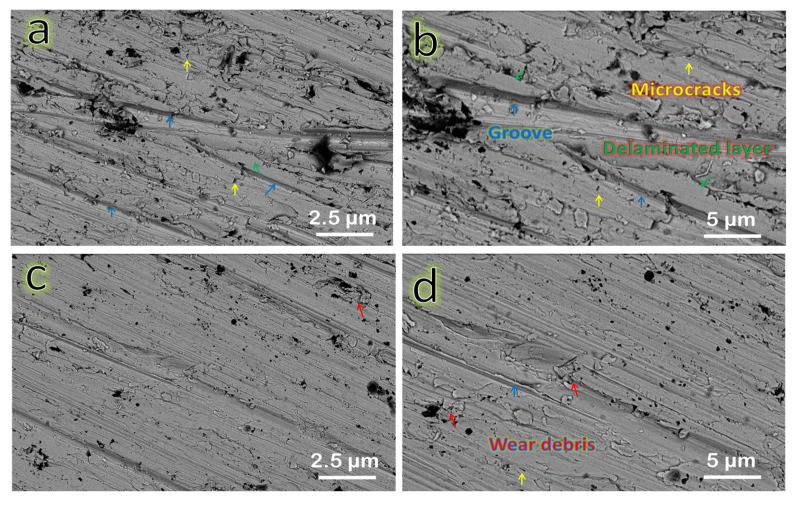
SEM images of wear tracks for the FNG0 sample at (**a**) lower magnification and (**b**) higher magnification and for the FNG8 sample at (**c**) lower magnification and (**d**) higher magnification powers.

**Table 1 nanomaterials-13-00537-t001:** Composition of Fe waste powder (wt.%).

Element	Fe	C	Mn	Al	P	Others
wt.%	99.78	0.06	0.04	0.03	0.02	0.07

**Table 2 nanomaterials-13-00537-t002:** Composition of granite waste powder (wt.%).

Element	SiO_2_	Al_2_O_3_	Fe_2_O_3_	CaO	MgO	K_2_O	TiO_2_	Residue
wt.%	62.89	18.19	6.19	4.97	2.64	3.42	1.58	0.12

**Table 3 nanomaterials-13-00537-t003:** Batch design of the prepared nanocomposites.

Sample Code		Composition (Vol.%)		
	**Fe**	**Cu**	**NbC**	**Granite**
**FNG0**	93	10	0	0
**FNG1**	91	10	0.5	1
**FNG2**	89	10	1	2
**FNG4**	85	10	2	4
**FNG8**	77	10	4	8

**Table 4 nanomaterials-13-00537-t004:** The effect of added hybrid reinforcements on the crystal size, lattice strain and dislocation density of the prepared samples.

Sample	Crystal Size (nm)	Lattice Strain (%)	Dislocation Density (%)
FNG0	32.47	0.2882	9.49 × 10^−4^
FNG1	30.26	0.3092	10.92 × 10^−4^
FNG2	27.18	0.3443	13.54 × 10^−4^
FNG4	22.77	0.4110	19.30 × 10^−4^
FNG8	17.82	0.5251	31.49 × 10^−4^

## Data Availability

The data presented in this study are available on request from the corresponding author.

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
