# Peer review of "Production of Hybrid Nanocomposites Based on Iron Waste Reinforced with Niobium Carbide/Granite Nanoparticles with Outstanding Strength and Wear Resistance for Use in Industrial Applications"

_nanomaterials, 2023, doi:10.3390/nano13030537_

Round 1
Reviewer 1 Report
This manuscript studied enhanced strength of iron waste/ niobium carbide/granite nanocomposites by microhardness method. Strictly speaking, hardness is not an accurate measurement of tensile strength of the material. If one wants to know the tensile strength of the composites, tensile bar methos is the only option. Hardness test is just an one dimensional measurement. Stress strain can show you strength and elongation (or toughness).
1. Are those wasted Fe material be purified or cleaned before experiment, It might contain oil during machining process.
2. L85 a force of 40×106 Pa, Pa is a pressure unit instead of force.
3. Results of Figure 11 should have STD (error bar) values.
4. Fig. 13. a) Weight loss and b) wear rate as a function of the sliding distance of the Fe 360. This title caption is not ringh.
5. Several typos. Should pay attention to this before submitting manuscripts.
a. L67 CHARACTERISTICS
B. L71 reinforcment
c. Figure 6 Particle L198 on the of crystal
d. L231 specimen s
e L232 h -> hour
f. Figure 8C Apparenit -> apparent
g. Line 408: Figure 15 (b) should be (c)
h. Line 397: Fig10 -> Fig. 15
Author Response
- Are those wasted Fe material be purified or cleaned before experiment, It might contain oil during machining process.
Answer: Thank you very much for your valuable comments which will add more value to our manuscript. The waste Fe was washed with high purity gasoline to remove the oil (if any) and then, absolute ethyl alcohol was used for the same reason.
- L85 a force of 40×106 Pa, Pa is a pressure unit instead of force.
Answer: The authors apologize for this inadvertent error, as the Pascal is actually a unit of pressure, and a force was written in error, and this has been corrected.
- Results of Figure 11 should have STD (error bar) values.
Answer: An error bar has been added to the mentioned figure
- Fig. 13. a) Weight loss and b) wear rate as a function of the sliding distance of the Fe 360. This title caption is not ringh.
Answer: Thankful for this correction, it has already been done
- Several typos. Should pay attention to this before submitting manuscripts.
- L67 CHARACTERISTICS
- L71 reinforcement
- Figure 6 Particle L198 on the of crystal
- L231 specimen s
e L232 h -> hour
- Figure 8C Apparenit -> apparent
- Line 408: Figure 15 (b) should be (c)
- Line 397: Fig10 -> Fig. 15
Answer: In response to your comment, the mentioned typos were corrected.
Reviewer 2 Report
In this work, the authors investigate the Recycling iron waste to produce hybrid nanocomposites reinforced with niobium carbide/granite nanoparticles with outstanding strength and wear resistance for use in industrial applications. The work is very interesting and with a high impact on literature. The manuscript can be accepted for the publication in Nanomaterials after addressing the following major comments.
Below I present the observations related to the content.
1. The title i not very suggestive. Please reformulate.
2. Abstract: Describe more relevant results in the abstract. Mention the purpose for which this study was conducted.
3. The lengthy sentences may be split in to smaller sentence without change of its meaning.
4. Also, suggested to include the recent references in the introduction part. In the introduction can be completed with properties and application of ferrite. The discussion can be extended by using the following papers: Acta Chimica Slovenica 56 (2), 379-385, Journal of thermal analysis and calorimetry 112 (1), 2013, 447-453
5. At the end of the introduction, the authors described the purpose (objectives) of the study, mentioning also the methods used, obviously briefly, but this should be explained in more detail in the methods.
6. The results and discussions part should be compared with the literature data. To redo the part of results and discussions by a systematic presentation of the results by which the readers of the articles manage to follow the article more easily.
7. Compare XRD results with other articles.
8. Calculate the XRD parameters and enter in a table. Then they should be compared with other literature studies. Do not use literature. Present the results obtained but do not compare with other existing studies. Are the values higher or lower than the literature? Add a comparative table.
9. Why is the size of the particles larger than the size of the crystallites? There is information in the literature.
10. TEM figures quality is poor throughout. To improve the quality of the figures. Enlarge the characters in the fgures.
11. TEM interpretation can be extended.
12. The same font sizes were not used in all the figures, according to the guide.
13. In the conclusions appear aspects that should appear in the chapter of results and be described only strictly the conclusions of this study. In case of conclusions, you could explain what does it add to the subject area compared to other published material?
14. References are not written according to the guide (not all authors are listed in all references, they are not marked with initials, the titles of the articles and journals are not mentioned in many places, the volumes or pages are missing).
15. Maybe the organization of the work would look better if you moved the chapter of magnetic measurements to the end of the work to make room for applications.
Author Response
- The title i not very suggestive. Please reformulate.
Answer: Thank you very much for your opinion which encourages us to do more in the future. In response to your valuable insight, the title has been changed to be more attractive to readers
- Abstract: Describe more relevant results in the abstract. Mention the purpose for which this study was conducted.
Answer: More relevant results have been added to the abstract. Also, the aim of the work was better clarified.
- The lengthy sentences may be split in to smaller sentence without change of its meaning.
Answer: This correction has been done.
- Also, suggested to include the recent references in the introduction part. In the introduction can be completed with properties and application of The discussion can be extended by using the following papers: Acta Chimica Slovenica 56 (2), 379-385, Journal of thermal analysis and calorimetry 112 (1), 2013, 447-453.
Answer: The discussion has been extended and the mentioned references have been added.
- At the end of the introduction, the authors described the purpose (objectives) of the study, mentioning also the methods used, obviously briefly, but this should be explained in more detail in the methods.
Answer: This modification has been done.
- The results and discussions part should be compared with the literature data. To redo the part of results and discussions by a systematic presentation of the results by which the readers of the articles manage to follow the article more easily.
Answer: Since our work is new, the literature did not contain work relevant to the samples examined. Therefore, some of the few data available in the relevant literature were used to compare some of the results obtained in this work.
- Compare XRD results with other articles.
Answer: Thank you for this comment. We would like to point out that the obtained XRD patterns are compared to standard cards, i.e. ICCD file cards which provide crucial evidence of the phases present in the samples. Also, the composition chosen for the samples in this work is new and therefore cannot be compared with the literature.
- Calculate the XRD parameters and enter in a table. Then they should be compared with other literature studies. Do not use literature. Present the results obtained but do not compare with other existing studies. Are the values higher or lower than the literature? Add a comparative table.
Answer: As discussed in the former point, the composition chosen for the samples in this work is new and therefore cannot be compared with the literature. Furthermore, the XRD parameters depend on the conditions chosen for the milling process and the types of ceramics used. Therefore, the obtained XRD is not comparable with the literature. However, it should be noted that the general concept is that the addition of ceramics to metal/metal alloys using the powder metallurgy technique results in a noticeable reduction in crystal size and an increase in other parameters.
- Why is the size of the particles larger than the size of the crystallites? There is information in the literature.
Answer: It is well-known that the particle consists of many crystals. Therefore, the particle size must be larger than the crystal size. It is worth noting that if the particle size is equal to the crystal size, it means that the formed material is a single crystal and of course it is a unique condition and requires many complex conditions to prepare.
- TEM figures quality is poor throughout. To improve the quality of the figures. Enlarge the characters in the fgures.
Answer: The authors are grateful for this amendment, and the quality of the images has been improved and the size of the characters has been enlarged
- TEM interpretation can be extended.
Answer: It has been explained in detail.
- The same font sizes were not used in all the figures, according to the guide.
Answer: The font sizes of all figures were unified.
- In the conclusions appear aspects that should appear in the chapter of results and be described only strictly the conclusions of this study. In case of conclusions, you could explain what does it add to the subject area compared to other published material?
Answer: This correction has been done
- References are not written according to the guide (not all authors are listed in all references, they are not marked with initials, the titles of the articles and journals are not mentioned in many places, the volumes or pages are missing).
Answer: According to your comment, the references were carefully revised and corrected.
- Maybe the organization of the work would look better if you moved the chapter of magnetic measurements to the end of the work to make room for applications.
Answer: Thank you for this comment. I would like to clarify that we did not conduct the magnetic properties because they are not related to the applications of the prepared nanocomposites; namely automobile brakes, gearbox and wind turbines.
Round 2
Reviewer 2 Report
Thank you for your efforts to improve the quality of the article. The authors took into account all my concerns. I think in the present form it can be accepted for publication.